# Diabetic Foot Disease during the COVID-19 Pandemic

**DOI:** 10.3390/medicina57020097

**Published:** 2021-01-22

**Authors:** Andrew J. M. Boulton

**Affiliations:** Manchester Diabetes Centre, Peter Mount Building, Manchester Royal Infirmary Manchester, Oxford Road, Manchester M13 9WL, UK; ABoulton@med.miami.edu; Tel.: +44-161-276-4452 (ext. 4406)

**Keywords:** diabetic foot, COVID-19 pandemic, foot ulceration, amputation

## Abstract

Throughout 2020, the COVID-19 pandemic has had a major impact on the care of non-communicable diseases across the world and diabetes is no exception. Whereas many branches of medicine have adapted to telemedicine, this is difficult and challenging for the diabetic foot which often requires “hands on” treatment. This review covers the challenges that have faced clinicians across the world in the management of complex diabetic foot problems and also includes some illustrative case vignettes which show how it is possible to manage foot ulcers without the usual access to laboratory and radiological testing. There is no doubt that the COVID-19 experience when handling diabetic foot problems will likely transform our approach to the management of diabetic foot disease especially in the areas of digital health and smart technology.

## 1. Introduction

### 1.1. The COVID-19 Pandemic

It was in late 2019 that a pneumonia of unknown cause was first reported from Wuhan, China: this was later named “coronavirus disease 2019” (COVID-19) by the World Health Organisation [1]. At the time of writing there have been >85 million COVID-19 cases across the world with more than 1.8 million fatalities. To date there have been >22 million cases in the USA and 2,500,000 in the United Kingdom.

It soon became apparent that older patients with other comorbidities, including diabetes, were at greater risk of poor outcomes and death [2,3]. One of the early studies from Wuhan focused on the relationship between diabetes and COVID-19 and reported that the risk of Intensive Care Unit admission and fatality was much higher in patients with pre-existing diabetes and COVID-19 [4].

Recognising the frequency of diabetes in patients with COVID-19, the International Diabetes Federation has been at the forefront of increasing the importance of good diabetes care to those people living with diabetes across the world [5]. During the last 12 months there has understandably been a swing back of research and media interest into communicable diseases such as COVID-19. At the height of the pandemic, this resulted in lesser attention being payed to non-communicable diseases such as heart disease, cancer, and diabetes, which remain the major cause of mortality across the world [6]. Thus, for many reasons, the COVID-19 pandemic has raised major challenges to the entire diabetes community [5,7]. It is therefore vital throughout this pandemic that the most vulnerable populations, including those with diabetes and obesity, will require support and the necessary resources to restrict a day-to-day exposure to infection [7].

### 1.2. Diabetic Foot Disease

Foot ulceration is the most common and costly late complication of diabetes, with morbidity and mortality being worse than for many cancers [8,9,10]. Data suggest that up to one in three of all people with diabetes will develop a diabetic foot ulcer (DFU) sometime during their lifetime. Non-healing DFUs are a leading cause of hospitalisation, amputation, disability, and death among the diabetic population [8].

Pathways to DFUs are well defined, and those at greater risk of developing such lesions are outlined in Table 1.

Two of the most important risk factors for DFU development are neuropathy and peripheral vascular disease. Due to loss of sensation in diabetic neuropathy, patients have poor appreciation of their DFU risk resulting in the lack of preventative foot self-care [11].

## 2. Diabetic Foot Disease during the COVID-19 Pandemic

According to one of the earlier papers from Wuhan, risk factors for mortality in those with diabetes who are admitted with COVID-19 include being elderly, being male with poor glycaemic control, hypertension, and cardiovascular disease [4]. As can be seen from Table 1, these risk factors for poor outcomes are also the very same risk factors for those people with diabetes who go on to develop foot complications [9]. The similarities of risk are summarised in Table 2.

The global pandemic has presented many challenges in the management of people with diabetes particularly with late complications such as risk factors for foot ulceration. New modes of patient consultation have widely been used during the pandemic including the use of telephone consultations and telemedicine sometimes with video consultation. Thus, the outpatient management of people with diabetes and its complications has faced a huge challenge during the last twelve months and in many countries, the classical “face-to-face” clinics have been cancelled and replaced by telephone consultations as noted above [12]. This development has therefore posed a threat to those with diabetic foot problems including active DFUs, ischaemia, and Charcot neuroarthropathy. The proper and careful clinical examination and assessment of risk of diabetic foot disease still requires a full physical examination of the lower limbs [9]. Realising the risk to people with neuropathy and vascular disease, the American Podiatric Medical Association put out a special communication in 2020 emphasising the importance of appropriate management of DFUs and other complications during these difficult times [13]. Other threats to those with neuropathy and peripheral arterial disease are listed in Table 3. As can be seen, during the strictest of lockdowns in many countries, most routine investigations which are normally required for the assessment of the diabetic foot, even including a plain X-ray, were not possible.

A further problem amongst people with diabetes has been an understandable and real element of fear. Widespread publicity of the COVID-19 pandemic has alerted those with diabetes to realise that should they be admitted to the hospital, the risk of a poor outcome is higher than in those without diabetes. Thus, many people with diabetes have been truly frightened to attend hospital clinics as they quite rightly perceive that the hospital is likely to be occupied by many patients with active COVID-19 disease. This perceived and in many ways realistic fear has led to a reduced attendance rate that has been observed in many hospital outpatient clinics (Boulton AJM, Personal Communication). Thus, in summary, the current pandemic of COVID-19 raises the possibility of the world experiencing a tsunami of late complications of diabetes in subsequent years when the pandemic is over.

The next section of this article will review specific reports on diabetic foot care from countries around the world during the last twelve months.

## 3. Diabetic Foot Problems during the COVID-19 Pandemic: Global Reports

China—it was in China that the coronavirus COVID-19 was first identified and there have been reports of difficulties with managing DFUs from China. One early communication from Hangzhou reported significant reductions in hospitalisations for diabetic foot problems during the first three months of 2020 [14]. Overall, the study that also looked at admissions later during 2020, suggested that the COVID-19 outbreak had a serious and disruptive effect on the delivery of hospital care for those with DFUs. These authors also raised the likelihood that patients feared attending the hospital for similar reasons to those outlined earlier in this review.

European Union—it was Italy that suffered amongst the worst of the consequences of the first wave of the COVID-19 pandemic in Europe leading to an almost complete lockdown of the country in the first quarter of 2020 [15]. As reported by Caruso et al. [16] its widespread lockdown significantly affected patients with chronic diseases including diabetes and particularly those with DFUs. This group from Naples reported that patients with diabetes admitted to a Tertiary Care Centre for DFU management had a 3-fold increased risk of amputation compared with figures from 2019. A further report from Rome described the development of a new triage pathway to manage patients with DFUs such that those with severely complicated lesions were urgently seen at the hospital outpatient service and admitted if necessary, whereas those with less complicated DFUs were managed by telemedicine after a brief outpatient evaluation. This study group included 151 patients seen since February 2020 and of these, only three required a major amputation. The authors concluded that this triage pathway provided adequate management of DFUs during the pandemic and there were no cases of hospital-acquired COVID-19 infections [17]. In Eastern Europe, Urbancic-Rovan reported on the experience of diabetic foot ulcers during the pandemic in the small country of Slovenia [18]. Severe logistical problems were encountered in caring for patients with DFUs such as lack of public transport and, again, fear of infection with COVID-19, keeping patients away from the outpatient services. It was possible partially to compensate for this non-attendance by using telephone and email consultations.

Turkey—a group of authors from Istanbul reported on their development of an algorithm to manage diabetic foot problems during the COVID-19 pandemic [19]. One important recommendation that they made was to request a CT thorax for a pre-operative screening in any DFU patients requiring surgery for the detection of possible undiagnosed COVID-19.

United Kingdom and United States—an early report on the effect of the pandemic on diabetic foot care from Manchester, United Kingdom and Los Angeles, USA compared and contrasted the approach and outcomes during the first lockdown period of March/April 2020 [20]. The threats to the management of DFUs during this difficult time are summarised in Table 3. It was during this first six-week lockdown that virtually all routine tests for outpatient services were suspended, emphasising the importance of good clinical medicine with careful history taking and, especially, examination of the feet. The inability to perform X-rays made clinical signs such as the “sausage toe” [21] and the positive “probe-to-bone test” [8,9] very important as clinical markers for the presence of underlying osteomyelitis. The inability to do routine X-rays during this period of time led to the clinical decision alone being made to diagnose and treat osteomyelitis. Two cases from Manchester illustrate the importance of these signs. Figure 1 shows in this first case a large neuropathic wound on the medial side of the hallux which did probe-to-bone, suggesting osteomyelitis. As this patient was first seen in the week prior to lockdown, an X-ray was possible confirming the presence of osteomyelitis as shown in Figure 1b. The decision to treat with oral broad-spectrum antibiotics was made and Figure 1c shows the healed lesion some eight weeks later. This showed healing and a subsequent X-ray confirmed radiologic healing.

Case 2 is illustrated in Figure 2 showing the clinical sign of a sausage toe (Figure 2a) and a plantar wound that actually probed deeply to bone, again suggesting osteomyelitis (Figure 2b). No X-ray was possible in this case as this was day 1 of lockdown; six weeks later after broad-spectrum oral antibiotics, the lesion was virtually healed, and the sausage toe was no longer so distended (Figure 2c).

In the cases reported from Los Angeles, the use of telemedicine was highlighted in [20]. Indeed, using a video connection, a home-visiting nurse was instructed how to apply and remove larval therapy (maggots) in a post-operative wound. Similarly, local amputation of toes was performed in an outpatient facility in a patient who was terrified to come to the hospital because of fear of contracting COVID-19: this patient had already had a renal transplant for end-stage renal disease and was therefore immuno-suppressed.

This tale of two cities confirms significant changes to the handling of diabetic foot problems including increased use of telephone and telemedicine contacts in both centres, and it was suggested that post-COVID-19 pandemic there may be new possibilities for models of care for the diabetic foot.

Latin America—Peru is the country that has been severely affected by COVID-19 infections in South America, and an observational study has reviewed how the pandemic has affected emergency visits to the Peruvian National Trauma Referral Centre in Lima [22]. They reported an 80% reduction of the number of severe patients seen in the emergency room in the first month of a lockdown. However, despite this overall decline in admissions, those with diabetic foot ulcers attending were increased in proportion to the total overall number of admissions. Indeed, during this period of lockdown, diabetic foot disease rose to the third most common indication for surgical admission, having been eleventh in the months prior. One possible reason for this observation was the reduction in non-COVID-19-related healthcare provision during this period.

Global—the International Diabetic Foot Care Group and Diabetic-Foot International have published a fast-track pathway for diabetic foot ulceration during the COVID-19 pandemic [23]. This pathway has been advocated as an easy tool for clinicians working in primary care and treating DFUs. It suggests that patients should be fast-tracked into three potential levels of severity and need for care: (a) uncomplicated diabetic foot ulcers; (b) complicated, defined as potentially ischaemic or infected with osteomyelitis; and (c) severely complicated DFUs, defined as gangrene, abscess, etc. This fast-track pathway and other proposals in this article provide a useful algorithm that can be adapted appropriately and used across the world. Naturally they propose early screening and investigation to ensure that patients do not in addition have COVID-19 infection.

## 4. Conclusions and the Future

In this Review, several approaches to the management of DFUs and more serious lower limb complications of diabetes have been reviewed from a number of different countries. The unifying and repeated message in all these reports has been not to neglect the diabetic foot and its potential complications during the pandemic. The art of clinical observation has never been more important in the management of diabetic foot disease. As the famous Irish physician, Dominic Corrigan, whose observations of aortic incompetence are now known as “Corrigan’s sign” wrote, “the trouble with most doctors is not that they do not know enough but they do not see enough.” A careful examination of the feet in people with diabetes is absolutely essential, and it always has been and always will be. However, as speculated by Najafi [24], the COVID-19 experience will likely transform our approach to the management of diabetic foot disease. There will undoubtedly be a new wave of innovations in the areas of digital health, smart technology including pressure or temperature sensing insoles, telehealth technologies, and more. Rogers et al. have entitled some of these changes “a Wound Center without walls” [25]. They propose the appropriate aggressive triage to provide care to patients with wounds across the spectrum of the health system not only using technology and community-centred care, but also, where needed, hospital provision. Most important in the management of diabetic foot disease is to provide the appropriate treatment to patients if possible while they are safely at home, but if not possible, to do so, where indicated, in outpatient facilities and hospital services for the most severely affected.

## Figures and Tables

**Figure 1 medicina-57-00097-f001:**
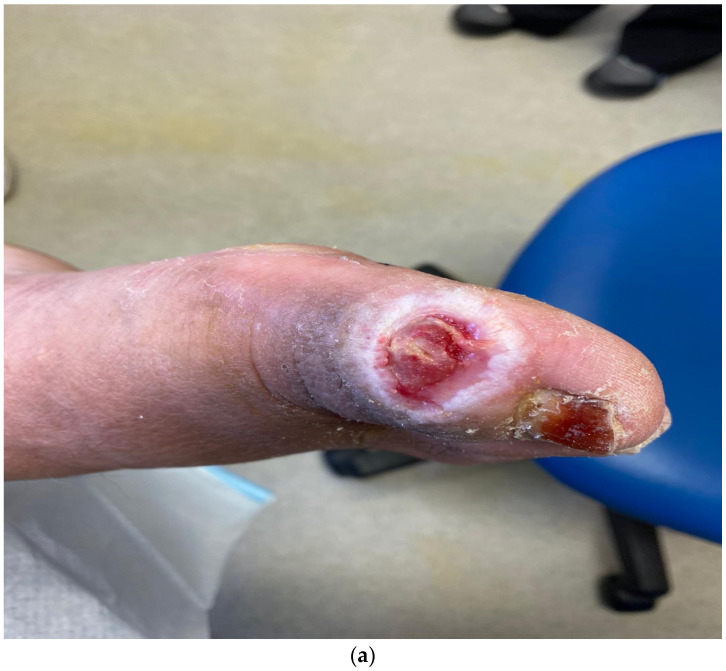
Images from case 1. (**a**) Foot ulcer on medial side of right hallux prior to lockdown. (**b**) Radiograph of right hallux prior to lockdown showing extensive osteomyelitis and septic arthritis in both phalanges and interphalangeal joints. (**c**) Healed wound on right hallux after 8 weeks of antibiotics, 6 weeks post-lockdown. (Reproduced from [20], with permission from ADA, 2020.)

**Figure 2 medicina-57-00097-f002:**
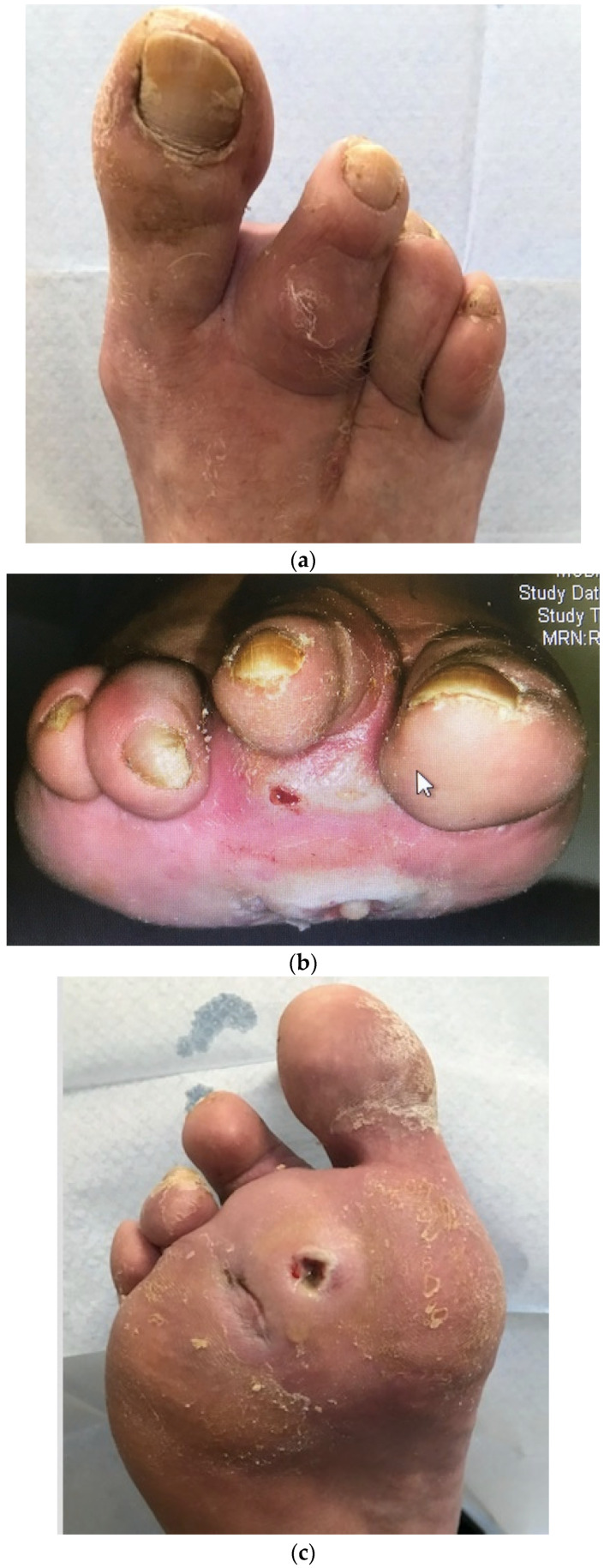
Images from case 2. (**a**) Dorsal view of the right foot showing sausage-shaped swelling of the second toe with tracking cellulitis (note previous amputation of third toe) when the patient first presented. (**b**) Frontal view of same toes showing purulent discharge from metatarsal head wound at presentation. (**c**) Plantar view of the right foot 6 weeks later, after continuous treatment with oral antibiotics showing near healing of the wound, reduced erythema, and reduced swelling of the toe. (Reproduced from [20], with permission from ADA, 2020.)

**Table 1 medicina-57-00097-t001:** Who is at risk of foot ulceration?

• Neuropathy
• Male sex
• Peripheral vascular disease
• Age
• Past history of foot ulceration
• Microvascular complications (especially nephropathy)
• Poor glycaemic control
• Cigarette smoking
• Foot deformity
• Amputation

**Table 2 medicina-57-00097-t002:** Similarities between those diabetic patients at risk of a poor outcome if hospitalised due to COVID-19 and those at risk of diabetic foot ulceration.

At-Risk Populations	
**For Mortality in those with Diabetes Admitted with COVID-19 in Wuhan [4]**	**For Diabetic Foot Ulceration [8,9]**
• Older	• Older
• Male	• Male
• Poor glycaemic control	• Poor glycaemic control
• Hypertension	• Hypertension
• CV Disease	• Other diabetic complications

CV = Cardiovascular.

**Table 3 medicina-57-00097-t003:** Diabetes and COVID-19: threats to the management of diabetic foot disease.

• Suspension of all routine lab tests, e.g., CRP, FBC, U&Es
• Suspension of all routine investigations, e.g., X-rays, MRI, and other imaging tests
• Suspension of routine non-invasive vascular laboratory tests
• Suspension of all non-emergency surgery including bypass (PAD and CAD) and minor amputations

Legend: CRP = c-reactive protein. FPC = full blood count. U&Es = urea and electrolytes. MRI = magnetic resonance imaging. PAD = peripheral arterial disease. CAD = coronary artery disease.

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
