# Peer review of "Diabetic Foot Disease during the COVID-19 Pandemic"

_medicina, 2021, doi:10.3390/medicina57020097_

Round 1

Reviewer 1 Report

The author have submitted and interesting revision of the current concern about how to manage diabetic foot patients during the COVID-19 pandemic. The manuscript is well written and gives a global perspective about how this concern has been faced and what could be happen in the near future. 

There is only a typo at line 56: it appears ccording and should appear According.

Author Response

Dear Reviewer 1,

Thank you for your review and your notes.

1.It has been revised

Reviewer 2 Report

In this manuscript, Boulton reviews diabetic foot disease in COVID-19 patients. In general the review is reasonable from a clinical point of view, though somewhat superficial. My feeling is that this may be a function of the amount of clinical information available. Nevertheless, the information is useful, especially in the context of treatment approaches for these patients in these challenging times. I have only minor comments below.  

Specific Comments

  1. Line 56; the “A is missing from “According.

  1. In Table 2, I would spell out the word cardiovascular as opposed to CV. Just one less abbreviation.

  1. At first glance, it is not clear what the ultimate value is of Table 3. I know that it is referenced below, but many of the items listed are similar/overlapping. It seems that this general information could be stated without a table.

  1. Figure 1b; you may want to use an arrow to point out the sites of osteomyelitis, which may not be as apparent to a non-clinical scientist reading the manuscript. In fact, the use of arrows for any of the clinical complications in the images that are being addressed in the text may be helpful.

Author Response

Dear Reviewer 2,

Thank you for your review and your notes.

1.It has been revised

2.Table 3 should remain (as a senior editor myself of a high impact fact journal I note that complex sentences are often better replaced by a small table)

3.Thinking that Medicina is a clinical journal, then clinicians would be able to recognise osteomyelitis and I honestly don’t think an arrow is necessary. Moreover as this is taken directly from a previous publication with permission, I really cannot change the picture so I suggest that it goes as is.

Reviewer 3 Report

This is a fascinating review about diabetic foot disease during the COVID-19 pandemic.

This paper is an important contribution and I recommend that it be accepted for publication.

Please check the sentence of Line 56 as it seems to be incomplete.

Thank you for the opportunity to review this very interesting manuscript.

Author Response

Dear Reviewer 1,

Thank you for your review and your notes.

 Line 56 has been revised